# Do Personality, Alcohol Abstinence Self-Efficacy, and Depressive Symptomatology Affect Abstinence Status in Treatment-Seeking Patients with Alcohol Use Disorder?

**DOI:** 10.3390/ijerph19159023

**Published:** 2022-07-25

**Authors:** Zofia Lebiecka, Ernest Tyburski, Tomasz Skoneczny, Jerzy Samochowiec, Adam Jędrzejewski, Jolanta Kucharska-Mazur

**Affiliations:** 1Department of Psychiatry, Pomeranian Medical University, 71-210 Szczecin, Poland; tskoneczny@gmail.com (T.S.); samoj@pum.edu.pl (J.S.); jola_kucharska@tlen.pl (J.K.-M.); 2Department of Health Psychology, Pomeranian Medical University, 71-460 Szczecin, Poland; ernest.tyburski@gmail.com (E.T.); adam.jedrzejewski3@gmail.com (A.J.)

**Keywords:** alcohol use disorders (AUDs), personality, depression, alcohol abstinence self-efficacy

## Abstract

Given the high global incidence and disabling nature of alcohol use disorders, alongside high relapse rates, we sought to investigate potential predictors of abstinence, considered a prerequisite of full remission. With an aim to examine (i) the effect of personality, alcohol abstinence self-efficacy, and depressive symptomatology on abstinence status as our primary objective, and (ii) interactions between these three factors, as well as (iii) their changes over time as two secondary objectives, we recruited 51 inpatients at an alcohol rehabilitation center to complete the International Personality Item Pool, the Alcohol Abstinence Self-Efficacy Scale and the Beck Depression Inventory, and to provide information on abstinence attainment 2 months post-treatment. Although regression analyses revealed no evidence for the effect of the investigated factors (personality, self-efficacy, or depressive symptoms) on post-therapy abstinence, other findings emerged, demonstrating (i) a significant reduction in the severity of depressive symptoms, (ii) the effect of personality and alcohol abstinence self-efficacy on depressive symptom severity, and (iii) the role of personality in predicting the temptation to use alcohol in recovering drinkers. These preliminary indications of links between personality, self-efficacy, and subjective well-being mark a promising area for future research on powerful and relevant cues of relapse and abstinence efficacy.

## 1. Introduction

Alcohol Use Disorders (AUDs) constitute highly prevalent and disabling disorders with mean lifetime incidence reaching approximately 8.6% in all countries combined [1]. A better grasp of etiological risk factors underlying AUDs could prove relevant to inform both prevention and intervention efforts [2], whose ultimate aim is that of recovery. Notwithstanding, there has been ongoing debate concerning the conceptualization of improvements that define what exactly is meant by “recovery” in persons afflicted with AUDs. Given the heterogeneity of methods in how patients reduce or resolve alcohol-related problems, the characterizations thereof have become increasingly more nuanced over time, encompassing factors such as improved well-being and daily functioning, beyond the mere attainment of abstinence or low-risk drinking [3]. However, a high relapse ratio and the extreme difficulty of remaining abstinent are still considered core characteristics of alcohol dependence [4], with relapse being a crucial event occurring in numerous individuals on their path towards recovery from AUD [5]. In this study, therefore, we sought to shed more light on the determinants and potential predictors of abstinence, considered a necessary condition and prerequisite of “sustained full remission” according to ICD-11 (lasting 1 to 12 months) [6]. In addition, while there is evidence supporting the predictive role of various psychosocial underpinnings of AUDs, factors related to abstinence/relapse outcomes seem to have been less explored.

There has been a long and ongoing debate on the effect of personality-related factors on alcohol misuse cf. [7,8]. Proposed as the most comprehensive conceptualization of personality, the Five Factor model, consisting of five higher-order personality traits, including openness to experience, conscientiousness, extraversion, agreeableness and neuroticism [9], is still amongst the most widely studied and applied in various conceptual frameworks. Considering this, available meta-analytic evidence suggests higher alcohol consumption in persons with low conscientiousness (responsible for self-control, flexibility, and adherence to social norms), low agreeableness (cooperativeness, kindness, and trust towards others), and high neuroticism (little emotional stability, proneness to anxiety, and sensitivity to negative emotions) [10]. Next to low agreeableness (compliance and straightforwardness in particular), low conscientiousness (mostly deliberation and dutifulness) and high neuroticism (i.e., impulsiveness and angry hostility), another meta-analysis seems to also implicate certain facets of extraversion (i.e., excitement-seeking) as likely predictors of pathological alcohol use [11], thus highlighting the associations between the two notions. Regarding facets related to alcohol abuse, past or present diagnoses thereof were found to be linked to low scores on “trust, achievement striving, self-discipline, and dutifulness” and high scores on “impulsiveness, vulnerability, and excitement-seeking” [12]. Of note, although dimension and facet results across populations may vary, the Five Factor framework does demonstrate common personality characteristics among AUD patients. Likewise, specific personality factors (psychoticism and persistence) and facets (novelty-seeking and impulsiveness) have been shown to serve as usable predictors not only for the very diagnosis of AUD, but also for the risk of relapse in alcohol-dependent samples [13], or to generally affect drinking outcomes [14].

Proposed as a key predictor of treatment outcomes and a likely mechanism underlying success in attaining abstinence or drinking reductions in response to treatment, is the psychological construct of self-efficacy [15]. Self-efficacy may be defined as one’s beliefs concerning their capacity to produce desired performance attainments [16], determining how much effort and energy they will put into changing an undesired behavior [16,17,18], and predictive of how persistent they will be in the case of initial failure. Of particular interest in alcohol use recovery is *abstinence self-efficacy*, i.e., a belief that an individual will be able to abstain from partaking in the undesired activity that is (excessive) drinking [19]. Given its theorized propensity to mediate behavior change and effective action, abstinence self-efficacy has been reported to also have an effect on alcohol use outcomes, such as predicting the time to first drink and the number of drinks on the first drinking day in individuals who fail to maintain abstinence after treatment [5]. There is evidence of higher self-efficacy scores in abstainers relative to those who relapsed [20] and findings suggesting that greater self-efficacy at discharge from treatment may predict abstinence [21,22,23]. There is also further evidence reaffirming the role of self-efficacy in predicting AUD outcomes [24], and findings to support links between self-efficacy and abstinence with increases in self-efficacy found to lower the likelihood of relapse [25].

In addition, there is a paucity of evidence on whether there are relationships between personality and self-efficacy in AUD patients, and, if so, whether those may be affected by comorbid depressive symptomatology. Given the high incidence of mood dysregulation in clinical samples [26], this may be considered a research priority. Depression is frequently associated with at-risk drinking cf. [27], both being highly prevalent phenomena, with approximately 280 million adults suffering from depressive disorders [28], 20% of whom meet the diagnostic criteria for AUD [29]. A role of negative emotionality underpinning addiction pathology [30], as well as associations between depressive symptomatology and failures in drinking control have been previously demonstrated [31]. Given the outbreak of the COVID-19 pandemic, whose aftermath has led to the severe deterioration of various aspects of mental health [32], including depression and substance use disorders, the urgency to identify the links between them warrants prompt investigation.

In light of the harmful effects of compulsive alcohol use on human functioning and the mitigated success of AUD-targeted therapies to date, with relapse rates for substance use disorders reaching 50% [33], the exploration of their likely underpinnings is both timely and valid. Should specific contributors to altering their outcomes be identified, treatment programs could be optimized to better suit individual patient needs. As described above, previous research found both factors (personality traits and facets and self-efficacy) to be individually linked to alcohol use and/or abstinence/relapse rates. However, no study has investigated how they may interact to affect the risk of relapse. To this end, we sought to examine (i) the effect of personality-related factors, alcohol self-efficacy, and depressive symptoms on attaining abstinence as our primary objective, and to analyze (ii) the potential interactions between the three investigated factors and (iii) their changes over time in a treatment-seeking AUD population as our two secondary objectives.

## 2. Methods

Fifty-one Polish male (*n* = 44; 86.3%) and female (*n* = 7; 13.7%) alcohol rehabilitation center inpatients, aged 25 to 62 years (*M* = 41.69, *SD* = 8.91), were enrolled to participate in an international study on the effect of modern technologies on enhancing treatment as usual (TAU) in the therapy of AUDs (for more information see [34]). As part of this project, upon screening for eligibility, they were subjected to a baseline interview with the use of the Mini-International Neuro-psychiatric Interview (MINI) for DSM-5 and invited to complete a questionnaire set, including the International Personality Item Pool (IPIP-50), the Alcohol Abstinence Self-Efficacy Scale (AASE) and the Beck Depression Inventory (BDI-II). Following TAU, at a two-month follow-up, forty-five participants (39 males and 6 females) completed a survey concerning post-treatment outcomes, including data regarding abstinence/relapse.

TAU was designed to last for 2 months and include mainly psychological CBT interventions, delivered via individual and group sessions. The treatment consisted of psycho-education, functional analysis of drinking situations, development and practice of coping strategies, problem-solving, and homework. Add-on interventions included tablet-assisted or virtual-reality-based approach/avoidance training programmes (AATP), each consisting of six sessions distributed over the period of 2 weeks and administered to 33 randomly selected participants (31 of whom provided information on abstinence at follow-up). The administered treatment had no effect on the results of the current study. Prior to enrolment, all patients provided their written informed consent to participate.

The eligibility criteria included: (1) written consent to participate in the study; (2) age ≥ 18 years; (3) completed detoxification (if needed); (4) no other SUDs; and (5) no severe psychiatric or neurological comorbidity or terminal somatic illness.

The baseline assessment was conducted with the use of the Mini-International Neuro-psychiatric Interview (MINI) for DSM-5, a structured interview assessing the 17 most prevalent psychiatric diagnoses through a set of yes/no questions [35,36].

A personality assessment was performed with the use of the 50-item International Personality Item Pool—a self-report personality questionnaire developed by Goldberg [37] to measure the Big Five personality traits, as expressed in Costa and McCrae’s [38] revised NEO personality inventory (NEO-PI-R). The tool is reported to have good validity and reliability (International Personality Item Pool, 2001).

Abstinence efficacy was assessed at two separate time points (baseline and discharge) with the 40-item Alcohol Abstinence Self-Efficacy Scale (AASE), consisting of subject ratings of the temptation to drink/confidence to abstain from alcohol use across 20 different high-risk situations. AASE was reported to demonstrate a solid subscale structure and strong reliability and validity. The scale comprises four five-item subscales labeled *negative affect*, *social positive*, *physical and other concerns*, and *withdrawal and urges*. The tool demonstrated no substantive gender differences [39].

Depressive symptoms were measured at two separate time points, i.e., baseline and discharge (pre- and post-treatment) with the Beck Depression Inventory (BDI-II), a 21-item self-report and a commonly applied psychometric scale enabling assessment of the severity of depression. The tool shows good reliability (Pearson *r* = 0.86) and internal consistency values (α = 0.91) [40].

## 3. Statistical Analysis

Statistical analysis was performed using the SPSS 27 and AMOS 7 (IBM Corp., Redmont, VA, USA). Continuous variables are presented as means (*M*) and standard deviations (*SD*). The normality of the distribution of continuous variables was tested with the Shapiro–Wilk test, kurtosis and skewness. Skewness between −2 and +2 and kurtosis between −7 and +7 were assumed to indicate normal distribution of variables [41]. Multivariate logistic regression (backward elimination, the Wald chi-square test) was further performed to identify independent factors (personality traits, dimensions of self-efficacy, and depressive symptoms) for abstinence (with the Hosmer–Lemeshow test for evaluating the goodness of fit of logistic regression models). The dependent sample Student’s *t*-test was further used to check the changes in depressive symptoms (before and after therapy). The effect size of the change in time was assessed with Cohen’s *d* [42]. The relationship between personality traits, dimensions of self-efficacy and depressive symptoms (at two time points) was checked using the Pearson *r* correlation coefficient. Furthermore, Structural Equation Modeling (SEM) was used to investigate the impact (multiple regression model). The selected indices were: the chi-square statistic (χ^2^), the root mean square error of approximation (RMSEA) [43], the goodness-of-fit index (GFI) [44], and the comparative fit index (CFI) [45]. The RMSEA of <0.06, 0.08–0.10, and >0.10 were considered to indicate good, adequate, and poor fit, respectively, and GFI and CFI of >0.90 were considered to indicate an acceptable fit [46]. We used a bootstrap maximum-likelihood estimation with 2000 samples. The alpha criterion level was set at 0.05 in all statistical analyses.

## 4. Results

### 4.1. Personality, Self-Efficacy, and Depressive Symptoms as Predictors of Alcohol Abstinence

As shown in Table 1, the performed multivariate logistic regression revealed that none of the analyzed factors (personality traits, dimensions of self-efficacy, or depressive symptoms) proved significant for abstinence in the investigated group.

### 4.2. Changes in Self-Efficacy and Depressive Symptoms before and after Therapy

As shown in Table 2, there was a significant reduction in the severity of depressive symptoms after therapy in the investigated group (*p* < 0.001). The effect size for the change was medium (*d* = 0.60) However, statistical analysis did not show significant changes in any dimensions of self-efficacy (in terms of either its main or subscales).

### 4.3. Personality, Self-Efficacy and Depressive Symptoms

As shown in Table 3, we observed significant positive correlations between extraversion and depressive symptoms before and after therapy (*p* = 0.018 and *p* = 0.003, respectively), and a negative correlation between agreeableness and depressive symptoms after therapy (*p* = 0.010). Moreover, there were significant negative correlations between emotional stability and social temptation before therapy (*p* = 0.030), and craving temptation before (*p* = 0.046) and after therapy (*p* = 0.018). Similarly, there was a significant negative correlation between intellect and social temptation after therapy (*p* = 0.032). No significant links emerged between other personality traits and dimensions of self-efficacy (main or subscales) or depressive symptomatology.

To test the effect of personality traits on depressive symptoms measured before and after therapy, we adopted a path analysis methodology within a SEM framework. Considering only significant correlations, we then decided to add selected paths between personality traits and depressive symptoms to the model. Based on the criteria recommended by Hu and Bentler [46], the model showed good fit to data (χ^2^ = 0.23 and *p* = 0.634; RMSEA = 0.00 and *p* = 0.654; GFI = 0.998; CFI = 1.000; see Figure 1).

Table 4 shows standardized regression weights for the effects of the personality traits on depressive symptoms before and after therapy. As shown, extraversion had an overall effect on depressive symptoms before therapy (β = 0.331; *p* = 0.017). In addition, extraversion and agreeableness had an overall effect on depressive symptoms after therapy, but were not significant (β = 0.310; *p* = 0.068 and β = −0.217; *p* = 0.134; respectively). The recorded values of predicted variance were 11% for depressive symptoms before therapy and 20% for depressive symptoms after therapy. In general, extraversion was an important predictor of depressive symptoms. That is, patients who scored higher on extraversion were likely to report more depressive symptoms.

We then used the SEM framework to test the effect of personality traits and dimensions of self-efficacy measured before and after therapy. Similarly, considering only significant correlations, we then decided to add selected paths between personality traits and dimensions of self-efficacy to the model. Based on the criteria recommended by Hu and Bentler [46], the model showed good fit to data (χ^2^ = 11.80 and *p* = 0.067; RMSEA = 0.14 (CL 90%: 0.00–0.26) and *p* = 0.105; GFI = 0.926; CFI = 0.956; see Figure 2).

Presented in Table 5 are standardized regression weights for the effects of personality traits on dimensions of self-efficacy before and after therapy. As shown, we found emotional stability to have an overall effect on social temptation and craving temptation before therapy, and craving temptation after therapy (β = −0.291; *p* = 0.069; β = −0.281; *p* = 0.079; β = −0.177; *p* = 0.024; respectively). In addition, intellect was found to have an overall effect on social temptation after therapy (β = −0.159; *p* = 0.038). Recorded values of predicted variance were 8% and 3% for social temptation before and after therapy, respectively, and 8% and 3% for craving temptation before and after therapy, respectively. In general, emotional stability was an important predictor of craving temptation after therapy. That is, patients who scored higher on emotional stability were likely to score lower on craving temptation after therapy, and participants who scored higher on intellect were likely to score lower on social temptation after therapy.

As shown in Table 6, we found significant positive correlations between AASE temptation and depressive symptoms before and after therapy (*p* < 0.001 and *p* = 0.027, respectively). Moreover, there were significant positive correlations between all dimensions of AASE temptation (negative affect, social, physical, and craving) and depressive symptoms before therapy (*p* < 0.001 for all variables) and between three dimensions of AASE temptation (negative affect, physical, and craving) and depressive symptoms after therapy (*p* = 0.031; *p* = 0.018; *p* = 0.032, respectively).

In the third step, we used the SEM framework to test the effect of self-efficacy dimensions on depressive symptoms measured before and after therapy. Similarly, considering only significant correlations, we then decided to add selected paths between self-efficacy and depressive symptomatology to the model. Based on the criteria recommended by Hu and Bentler [46], the model showed good fit to data (χ^2^ = 0.27 and *p* = 0.872; RMSEA = 0.00 (CL 90%: 0.00–0.14) and *p* = 0.887; GFI = 0.997; CFI = 1.000; see Figure 3).

Presented in Table 7 are standardized regression weights for the effects of dimensions of self-efficacy on depressive symptoms before and after therapy. As shown, we found temptation to have an overall effect on depressive symptoms before and after therapy (β = 0.576; *p* = 0.002 and β = 0.306; *p* = 0.009). Recorded values of predicted variance were 33% and 9% for depressive symptoms before and after therapy, respectively. In general, temptation was an important predictor of depression symptoms before and after therapy. That is, patients who scored higher on AASE temptation were likely to score higher on depressive symptoms before and after therapy, but this effect was stronger before therapy.

Due to the small sample size, it was not possible to use the SEM model to estimate the relationship between separate dimensions of self-efficacy (negative affect, social, physical, and craving) and depressive symptoms before and after therapy.

## 5. Discussion

In this study, we investigated the postulated effect of personality-related factors, alcohol abstinence self-efficacy, and depressive symptomatology on post-therapy abstinence attainment in a treatment-seeking AUD population.

Against our initial expectations, none of the analyzed factors, i.e., personality traits, facets of alcohol abstinence self-efficacy, or depressive symptoms proved individually significant for abstinence, as reported by our sample at the two-month follow-up. This means that personality profiles, although seemingly significant for the development of AUDs c.f. [10,11], do not necessarily need to predict primary treatment outcomes in terms of relapse/abstinence attainment. Although there is evidence for the links between higher order personality factors (i.e., high neuroticism and low conscientiousness) and relapse rates at a 12 months post-treatment follow-up, postulated to remain in line with previous findings [47], and thus supporting the use of the Five Factor framework in alcohol-dependent cohorts, such findings are not reported across all research in AUD samples. Hence, elsewhere, Foulds et al. [48] demonstrate that the use of personality instruments other than Cloninger’s Temperament and Personality Questionnaire (TPQ) or Temperament and Character Inventory (TCI), may lack consistency in reporting findings on the links between personality variables and treatment outcomes. This remains in line with the results of Arnau et al. [49], who report associations between TCI scales, adherence to treatment, and relapse rates in alcoholic outpatients. In their study on the effect of personality on treatment outcomes, Muller et al. [13] used five different personality assessment tools (NEO 5-Factor Inventory, Temperament and Character Inventory, Eysenck Personality Questionnaire, Eysenck Impulsiveness–Venturesomeness–Empathy Scale and Sensation-Seeking Scale) to find that relapsed patients scored higher in psychoticism, novelty seeking, and impulsiveness, and lower in persistence, none of which are described within the Five Factor framework. This, alongside our results, raises an important question as to whether the use of instruments measuring personality factors according to the Five Factor theory is warranted in alcohol-dependent samples, or perhaps other theoretical models and corresponding tools are a better alternative in both research and practical approaches to predicting treatment outcomes. It can also be noted that as personality is known to affect behavior in a variety of ways, and the effect of personality-related factors on treatment outcomes can occur through mediation, the effect of potential mediating variables is definitely worthy of further investigation.

Likewise, and against our expectations or available evidence cf. [50], our results suggest no links between alcohol abstinence self-efficacy and abstinence/relapse rates. This suggests that the set of beliefs our patients hold concerning their capacity to refrain from drinking did not relate to their actual ability to do so as manifested behaviorally post-treatment. One way to interpret such a result could be that a likely phenomenon prevalent across our sample is low self-awareness reflected in the limited ability to generate a realistic self-report. More specifically, the participants might be prone to construing a particular self-image that is in fact an illusion of being capable of coping with emerging urges, which is to serve as a means to maintain their fragile self-esteem. This, in turn, could imply deficits within metacognitive processes, i.e., little awareness of their actual behavioral tendencies and/or an unrealistic representation of their true functioning c.f. [51]. Such meta-cognitive bias could be another way to account for the observed lack of consistency within self-reported vs experimental outcomes.

Similarly, contrary to existing research suggesting that depression symptoms are a significant predictor of addiction treatment outcomes [52,53,54,55], our findings indicate no relationship between depressive symptomatology and post-therapy abstinence status. (Sub)clinical depression symptoms can develop following acute and prolonged withdrawal from chronic substance misuse. Whether alcohol use precedes depressive symptoms or users reach for it to “self-medicate” and thus alleviate pre-existing depression symptoms, the inability to attain abstinence appears to be significantly connected with depression symptoms [56]. In our sample, however, the self-reported level of depressive symptoms/subjective emotional distress recorded either at the beginning of or after treatment did not affect whether or not they were going to be capable of controlling the temptation to drink emerging at any time between discharge and a two-month follow-up. This may therefore indicate the potential existence of other factors confounding the elsewhere-postulated relationship between depressive symptoms and abstinence/relapse rates, which makes controlling for other demographic and/or treatment-related variables a good research direction in the future.

Apart from investigating the effect of personality, alcohol self-efficacy, and depressive symptomatology on post-treatment ability to abstain from alcohol use in recovering AUD patients, we also sought to analyze the potential interactions between them, as well as their changes in time as two secondary objectives of the current study. Hence, even though we found no meaningful evidence for the effect of either one of the three investigated factors on post-therapy abstinence, certain other findings are definitely worthy of expanding on.

Perhaps somewhat surprisingly, we found extraversion to be significantly linked with depressive symptoms before and after therapy, and to predict their level at admission. In addition, depressive symptomatology at discharge was also negatively correlated with agreeableness. Given that previous findings tended to consider extraversion to be a factor that is generally protective against depression, highlighting its strong negative links to depressive symptoms and diagnoses [57,58], our results seem all the more unexpected. Nevertheless, one way they can be accounted for is through the consideration of the fact that more extraverted and socially-inclined individuals may experience more profound negative consequences of social isolation and other restrictions inpatient treatment is known to entail. Secondary outcomes of social distancing may assume the form of negative emotional states such as feelings of loneliness, frustration, concern, or boredom. These, however, do not need to affect all patients in a universal manner. Indeed, it has been proposed that extraversion, a factor involving sociability, assertiveness and high energy levels [9,59], could moderate the relationship between measure stringency and mental health problems e.g., [59,60,61,62,63]. Hence, although research to date is scarce and yields conflicting results [64,65,66,67], there are likely disadvantages to being an extravert in situations where severe restrictions and social distancing are in place (or even anticipated), as they inhibit their natural urges for social engagement [68], pleasure and excitement [69], or new and exciting surroundings [70]. Incidentally, our findings are in keeping with research which demonstrates that extraversion might moderate the relationship between measure stringency and depressive symptoms [71], as the constraints of an inpatient alcohol rehabilitation treatment may in many ways resemble those imposed during the COVID-19 pandemic. In turn, we found greater agreeableness to constitute a protective factor against depressive symptoms at discharge from alcohol treatment, indicating that individuals who are known to be more considerate, friendly, generous, helpful, and to value cooperation and social harmony tended to be less prone to reporting negative affectivity. Such an inverse relationship between self-reported depressed mood and agreeable personality disposition remains in line with previous findings across different cohorts [72,73,74,75,76] and suggests a potentially greater capacity to cope with the restrictions of an inpatient treatment while preserving a state of subjective well-being due to more proficient social skills.

Interestingly, we further revealed significant negative correlations between emotional stability and (i) social/positive factor of AASE temptation at baseline, (ii) craving and urges of AASE temptation at baseline and endpoint, and intellect and endpoint measurement of social/positive factor of AASE temptation, with further analyses suggesting that intellect may in fact constitute a significant predictor thereof, while emotional stability may significantly predict AASE craving and urges at discharge. Such results suggest that higher emotional stability correlates with a lesser-reported tendency to feel tempted to reach for alcohol during social engagements or to boost the subjective state of well-being (i.e., *seeing others drinking at a bar or at a party; excited or celebrating with others; when on vacation and wanting to relax; when people they used to drink with encourage them to drink; when offered a drink in a social situation*). Of note, however, such an association was only observed at baseline and lost towards discharge. This may mean that an individual disposition to cope with stressful situations in a calm and composed fashion could mitigate the anticipated external pressure to drink or the internal desire to use alcohol as a means to alleviate negative affective states during admission, i.e., at the time when AUD patients are motivated and full of hope that prospective therapy will result in favorable outcomes, and when their actual ability to reach for alcohol (due to imposed restrictions) seems remote. Towards discharge, however, their appraisal of the situation as well as their own feelings may change to become more realistic, especially when the opportunity to use becomes so much more tangible. In turn, and as may have been expected, the negative correlation between emotional stability and craving and urges persists throughout therapy, suggesting that the disposition not to feel tempted even in the case of suffering from withdrawal, craving, or tested willpower (i.e., *being in agony because of stopping or withdrawing from alcohol use; having the urge to try just one drink to see what happens; feeling a physical need or craving for alcohol; wanting to test willpower over drinking or experiencing an urge or impulse to take a drink when caught unprepared*) may be more stable in individuals who are also more stable emotionally. The links we found between individual differences and alcohol abstinence self-efficacy also suggested that higher intellect scores may predict a lesser inclination to feel tempted to drink in social situations or those where alcohol is used to enhance positive emotionality. Intellect, equivalent to openness to experience, reflects an inherent tendency to enjoy novelty, variety, and change, and is frequently found in curious, imaginative, and creative individuals. Such greater mental capacity for reflection may therefore constitute a prerequisite for better cognitive (rather than impulsive affect-dominated) processing of cravings resulting in a greater ability to control/inhibit automatic approach tendencies aimed at indulging emergent urges. Nevertheless, given that both scores (i.e., intellect and the social/positive aspect of AASE) represent self-reported beliefs our participants hold about themselves, they may not necessarily reflect their real disposition towards the good cognitive control of the temptation to drink in social engagements, but may instead indicate how they desire to see themselves. Alternatively, such results could be construed to indicate that other factors (beyond the experienced temptation to drink) might moderate their behavior, as indeed, the reported greater capacity to cope with urges apparent in social context did not translate to either greater confidence they would refrain from drinking or higher abstinence results at follow-up predicted by either of the two measures.

Of note, significant links emerged between alcohol abstinence self-efficacy and depression. Given the temporal stability of alcohol abstinence self-efficacy in time as suggested by our results (cf. no significant changes in AASE scores vs changes within BDI scores pre- vs post-treatment, as presented in Table 2), we no longer considered alcohol abstinence self-efficacy and depressive symptomatology as interdependent variables, but rather assumed the former to predict the level of the latter both before and after therapy, and therefore estimated alcohol abstinence self-efficacy’s effect on depression level using the SEM framework. Such an approach resulted in demonstrating positive correlations between AASE temptation and depressive symptomatology at both investigated time points (pre- and post-therapy). More specifically, the observed associations included those between all the facets of the former (i.e., negative affect, social/positive, physical and other concerns and craving and other urges) and depressive symptom severity at baseline and endpoint, with initial correlation with social/positive no longer significant at discharge. Self-efficacy is known to play a positive role in alleviating depressive symptoms, as efforts to build and maintain a sense of control over one’s life and environment might serve to dynamically reappraise stressful circumstances and establish a certain resistance to depressive symptoms [77,78]. A strong sense thereof is further known to enhance personal achievements, reduce stress, and lower vulnerability to depression. In turn, alongside greater vulnerability to stress and depression [77,79,80], low self-efficacy might contribute to doubting one’s own capacities, formulating low aspirations, and generally manifesting poor commitment to established goals [81]. Separate, yet closely related to efficacy, temptation may be construed as a measure of cue strength and its capacity to precipitate alcohol use. Hence, the temptation to reach for alcohol in a particular context can be low and the efficacy to abstain quite high. This, however, may not always be the case, especially during recovery when, in spite of high temptation, an individual’s levels of efficacy to abstain could still remain high due to acquired skills, motivation and commitment [82]. In our sample, temptation proved to correlate with the symptoms of depression throughout the study, meaning that a higher-experienced desire to drink was linked with a higher depression level. This could be interpreted in two different ways. AUD patients could construe the high-experienced temptation to use alcohol as surpassing their ability to resist and thus consider it a significant predictor of post-therapy relapse and/or poor prognostic of therapeutic success. Such an appraisal, in line with the theoretical approach proposed by DiClemente et al. [39], suggesting an inverse relationship between temptation and self-efficacy, could result in a more pessimistic outlook on their actual ability to abstain from drinking and consequently a more depressive mood. Conversely, a more depressive mood could entail a more negative evaluation of one’s own tendencies to consider various cues and situations in terms of a temptation or a cognitive inclination towards making such types of appraisals.

As previously indicated, another interesting finding was a significant drop in depressive symptoms recorded at discharge. Managing depression symptoms and regulating emotional distress are not often prioritized in normal substance misuse treatment [83,84]. Rather, the conventional treatment focus is on measures to promote abstinence and decrease substance use. The available research shows that therapies aimed at reducing the negative consequences of depression symptoms and subjective distress (e.g., susceptibility to stress-induced and cue-induced craving) could help improve treatment success. Our results, indicating a significant drop in depressive symptomatology, could mean that the administered interventions or other treatment-related variables had an overall positive effect on the self-reported psychological well-being of the participants and could thus lead to improved treatment outcomes. This shows that pharmaceutical and behavioral methods aimed at regulating negative emotionality or alleviating depressive symptoms require particular consideration in conventional substance use treatment. Furthermore, a continuity of care approach for depression and depressive symptoms following the cessation of addiction treatment may be required for many AUD individuals in order to maintain attained alcohol-free status [85]. Perhaps it could therefore be the varied post-discharge management of mental health across our sample that might potentially explain the unexpected lack of associations between the recorded levels of depression and abstinence status. In addition, in regards to our patients’ self-efficacy, the reduction in depressive symptomatology post-therapy could have a boosting effect on cognitive performance and potentially underlie a more favorable approach to self-efficacy (and temptation) appraisals, thus deeming the initially observed link between depression and social/positive facets of AASE temptation ultimately insignificant.

## 6. Limitations and Future Directions

Certain limitations of this study point to promising future research directions. Notably, our findings were based on a relatively small and homogeneous (also in terms of gender) sample of treatment-seeking rehabilitation center inpatients, which could contribute to a specific personality profile. Future research could look into replicating it in larger alcohol-use populations with more variation in demographics and drinking outcomes, and with an extended time of observation. Similarly, while self-report tools appear to capture the subjective aspect of individual functioning, they lack the objectivity provided by experimental paradigms or informant ratings. Future research efforts could be expanded to boost ecological validity and shift away from traditional clinical settings and toward virtual-reality-based trials in an effort to reduce reliance on self-report measures. Taking into account the entire scope of our findings and the evidence from previous research [74] demonstrating both the links between Five Factor personality traits and depressive symptoms, as well as the relationship between self-efficacy and depressive symptoms, a possibility should be considered that the effect of personality on depressive symptoms could be mediated through the effect of personality on self-efficacy. Quite remarkably therefore, our results open a new direction for future research on the role of self-efficacy as a mediator between personality and depressive symptoms in alcohol-dependent cohorts. This becomes particularly significant taking into consideration that understanding the role of self-efficacy in addictive behavior can provide valuable information, especially in terms of abstinence attainment, for efficacy evaluations are viewed as a critical component of the relapse crisis [86].

## 7. Conclusions

In an attempt to identify predictors of abstinence in treatment-seeking AUD patients, we found that none of the three selected factors, i.e., personality, alcohol abstinence self-efficacy, or depressive symptomatology, had an expected effect on post-therapy abstinence at a two-month follow-up. Nevertheless, this longitudinal study has led to certain valuable findings. Specifically, personality and alcohol abstinence self-efficacy proved to have an effect on subjective well-being, as reflected in self-reported depressive symptom severity. Furthermore, although not found to translate to actual abstinence rates, mostly due to the fact that the links between temptation and self-efficacy are considered quite complex, certain aspects of personality were also likely to predict the temptation to use alcohol in recovering drinkers. These preliminary indications of relationships between personality, self-efficacy, and subjective well-being mark a promising area for the search for powerful and relevant cues of abstinence efficacy as well as relapse, thus outlining a fertile field for future research.

## Figures and Tables

**Figure 1 ijerph-19-09023-f001:**
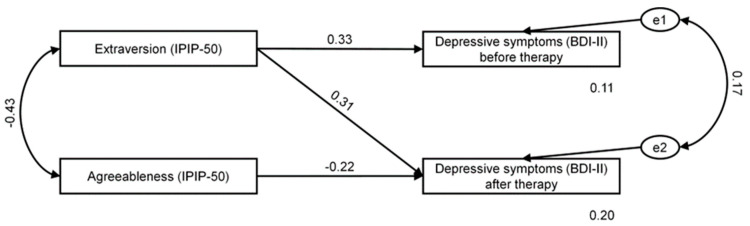
Extraversion and agreeableness as predictors of depressive symptoms in the investigated group before (T1BDI) and after (T2BDI) therapy. e1 and e2 depict the error term referring to the dependent variables.

**Figure 2 ijerph-19-09023-f002:**
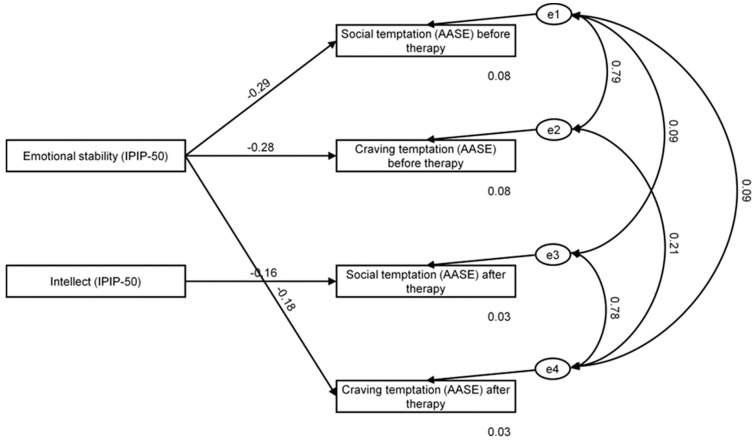
Emotional stability and intellect as predictors of self-efficacy in the investigated group before and after therapy. e1–e4 depict the error term referring to the dependent variables.

**Figure 3 ijerph-19-09023-f003:**
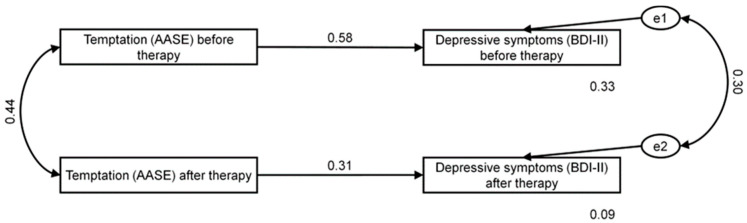
AASE temptation as predictor of depressive symptoms in the investigated group before and after therapy. e1 and e2 depict the error term referring to the dependent variables.

**Table 1 ijerph-19-09023-t001:** Personality traits, dimensions of self-esteem, and depressive symptoms as predictors of abstinence in the investigated group.

Variable	Odds Ratio	95% Confidence Interval	*p*	χ^2^	*p*
Personality traits in IPIP-50 as predictors					
Extraversion	1.04	0.92–1.18	0.541	2.98	0.702
Agreeableness	1.07	0.92–1.24	0.388		
Conscientiousness	1.05	0.93–1.18	0.462		
Emotional stability	1.07	0.89–1.27	0.484		
Intellect	0.87	0.70–1.09	0.224		
Dimensions of alcohol abstinence self-efficacy (main scale scores) as predictors					
Temptation	1.00	0.97–1.03	0.925	0.03	0.988
Confidence	1.00	0.97–1.03	0.899		
Dimensions of alcohol abstinence self-efficacy (temptation subscale scores) as predictors					
Negative affect temptation	1.14	0.86–1.52	0.369	4.34	0.362
Social temptation	0.80	0.62–1.04	0.099		
Physical temptation	0.87	0.67–1.13	0.289		
Craving temptation	1.23	0.91–1.67	0.181		
Dimensions of alcohol abstinence self-efficacy (confidence subscale scores) as predictors					
Physical confidence	0.94	0.76–1.17	0.592	1.43	0.840
Negative affect confidence	1.02	0.77–1.35	0.874		
Social confidence	0.90	0.72–1.14	0.378		
Craving confidence	1.16	0.83–1.62	0.377		
Depressive symptoms in BDI-II as a predictor					
Depressive symptoms	0.98	0.91–1.06	0.677	0.18	0.674

AASE = Alcohol Abstinence Self-Efficacy Scale. BDI-II = Beck Depression Inventory Second Edition. IPIP-50 = International Personality Item Pool.

**Table 2 ijerph-19-09023-t002:** Changes in dimensions of self-efficacy and depressive symptoms in the investigated group before and after therapy.

Variable	Scores before Therapy *M* (*SD*)	Scores after Therapy *M* (*SD*)	*t*	*p*	*d*
Changes in dimensions of alcohol abstinence self-efficacy (main scale scores)					
Temptation	30.04 (19.11)	25.69 (18.48)	1.57	0.123	-
Confidence	34.53 (23.67)	32.24 (27.61)	0.50	0.622	-
Changes in dimensions of alcohol abstinence self-efficacy (temptation subscale scores)					
Negative affect temptation	8.94 (5.23)	7.43 (5.53)	1.77	0.083	-
Social temptation	7.73 (5.37)	6.51 (4.94)	1.54	0.131	-
Physical temptation	5.75 (5.07)	5.16 (4.54)	0.87	0.389	-
Craving temptation	7.63 (5.04)	6.59 (4.63)	1.50	0.140	-
Changes in dimensions of alcohol abstinence self-efficacy (confidence subscale scores)					
Physical confidence	9.37 (5.75)	8.14 (6.66)	1.11	0.274	-
Negative affect confidence	8.20 (6.23)	8.02 (6.97)	0.15	0.883	-
Social confidence	8.43 (6.79)	7.94 (7.66)	0.39	0.701	-
Craving confidence	8.53 (6.21)	8.14 (7.02)	0.32	0.754	-
Changes in depressive symptoms in BDI-II					
Depressive symptoms	11.63 (8.63)	6.41 (5.25)	4.27	0.000	0.60

AASE = Alcohol Abstinence Self-Efficacy Scale. BDI-II = Beck Depression Inventory Second Edition.

**Table 3 ijerph-19-09023-t003:** Relationships between personality traits, dimensions of self-efficacy, and depressive symptoms in the investigated group before and after therapy.

**Variable**	**Dimensions of Alcohol Abstinence Self-Efficacy (Main Scale Scores)**										**Depressive Symptoms in BDI-II before Therapy**			**Depressive Symptoms in BDI-II after therapy**		
	**Temptation before Therapy**		**Temptation after Therapy**			**Confidence before Therapy**			**Confidence after Therapy**							
**Personality Traits in IPIP-50**	** *r* **	** *p* **	** *r* **		** *p* **	** *r* **	** *p* **		** *r* **	** *p* **	** *r* **		** *p* **	** *r* **	** *p* **	
Extraversion	0.14	0.323	0.16		0.253	0.03	0.848		−0.10	0.470	0.33		0.018	0.40	0.003	
Agreeableness	−0.04	0.773	−0.15		0.299	0.00	0.997		0.01	0.952	−0.20		0.162	−0.36	0.010	
Conscientiousness	−0.20	0.169	0.14		0.330	−0.10	0.469		0.08	0.592	−0.04		0.759	0.09	0.541	
Emotional stability	−0.25	0.081	−0.26		0.063	−0.04	0.785		−0.20	0.155	−0.23		0.107	−0.05	0.733	
Intellect	−0.23	0.108	−0.23		0.098	−0.13	0.354		−0.11	0.430	−0.09		0.513	0.11	0.463	
	**Dimensions of Self-Esteem (in Subscales of Temptation) in AASE**															
	**Before therapy**								**After therapy**							
	**Negative Affect Temptation**		**Social Temptation**		**Physical Temptation**		**Craving Temptation**		**Negative Affect Temptation**		**Social Temptation**		**Physical Temptation**		**Craving Temptation**	
**Personality Traits in IPIP-50**	** *r* **	** *p* **	** *r* **	** *p* **	** *r* **	** *p* **	** *r* **	** *p* **	** *r* **	** *p* **	** *r* **	** *p* **	** *r* **	** *p* **	** *r* **	** *p* **
Extraversion	0.22	0.115	−0.04	0.788	0.23	0.107	0.12	0.420	0.25	0.080	−0.03	0.809	0.24	0.087	0.15	0.280
Agreeableness	−0.14	0.319	0.04	0.781	−0.08	0.598	0.02	0.865	−0.20	0.159	−0.04	0.757	−0.21	0.140	−0.10	0.481
Conscientiousness	−0.21	0.149	−0.18	0.203	−0.22	0.127	−0.12	0.411	0.10	0.500	0.15	0.295	0.12	0.411	0.17	0.247
Emotional stability	−0.17	0.220	−0.30	0.030	−0.15	0.293	−0.28	0.046	−0.27	0.057	−0.24	0.089	−0.14	0.316	−0.33	0.018
Intellect	−0.15	0.309	−0.25	0.078	−0.22	0.117	−0.23	0.112	−0.17	0.229	−0.30	0.032	−0.23	0.101	−0.18	0.197
	**Dimensions of Self-Esteem (in Subscales of Confidence) in AASE**															
	**Before Therapy**								**After therapy**							
	**Physical Confidence**		**Negative Affect Confidence**		**Social Confidence**		**Craving Confidence**		**Physical Confidence**		**Negative Affect Confidence**		**Social Confidence**		**Craving Confidence**	
**Personality Traits in IPIP-50**	** *r* **	** *p* **	** *r* **	** *p* **	** *r* **	** *p* **	** *r* **	** *p* **	** *r* **	** *p* **	** *r* **	** *p* **	** *r* **	** *p* **	** *r* **	** *p* **
Extraversion	0.05	0.728	−0.05	0.710	0.03	0.824	0.08	0.590	−0.08	0.555	−0.11	0.436	−0.11	0.439	−0.10	0.507
Agreeableness	−0.01	0.968	−0.03	0.816	0.09	0.538	−0.06	0.678	0.03	0.836	−0.02	0.868	0.03	0.815	−0.01	0.959
Conscientiousness	−0.04	0.760	−0.07	0.603	−0.12	0.404	−0.15	0.297	0.07	0.611	0.08	0.600	0.08	0.597	0.08	0.596
Emotional stability	0.03	0.859	−0.06	0.700	−0.06	0.689	−0.05	0.703	−0.20	0.167	−0.21	0.147	−0.18	0.211	−0.21	0.141
Intellect	−0.15	0.305	−0.06	0.656	−0.09	0.528	−0.21	0.146	−0.14	0.319	−0.13	0.374	−0.05	0.709	−0.12	0.385

AASE = Alcohol Abstinence Self-Efficacy Scale. BDI-II = Beck Depression Inventory Second Edition. IPIP-50 = International Personality Item Pool.

**Table 4 ijerph-19-09023-t004:** Standardized regression weights for associations between personality and depressive symptoms before and after therapy in the investigated group.

	Estimate	*p*	Lower	Upper
Extraversion (IPIP-50)—Depressive symptoms (BDI-II) before therapy	0.331	0.017	0.070	0.542
Extraversion (IPIP-50)—Depressive symptoms (BDI-II) after therapy	0.310	0.068	−0.025	0.666
Agreeableness (IPIP-50)—Depressive symptoms (BDI-II) before therapy	−0.217	0.134	−0.511	0.063

BDI-II = Beck Depression Inventory Second Edition. IPIP-50 = International Personality Item Pool.

**Table 5 ijerph-19-09023-t005:** Standardized regression weights for associations between personality and self-efficacy in the investigated group before and after therapy.

	Estimate	*p*	Lower	Upper
Emotional stability (IPIP-50)—Social temptation (AASE) before therapy	−0.291	0.069	−0.571	0.018
Emotional stability (IPIP-50)—Craving temptation (AASE) before therapy	−0.281	0.079	−0.547	0.038
Emotional stability (IPIP-50)—Craving temptation (AASE) after therapy	−0.177	0.024	−0.336	−0.015
Intellect (IPIP-50)—Social symptoms (BDI-II) after therapy	−0.159	0.038	−0.303	−0.005

AASE = Alcohol Abstinence Self-Efficacy Scale. BDI-II = Beck Depression Inventory Second Edition.

**Table 6 ijerph-19-09023-t006:** Relationships between dimensions of self-efficacy and depressive symptoms in the investigated group before and after therapy.

	**Depressive Symptoms in BDI-II before Therapy**	
**Dimensions of Self-Esteem in AASE before Therapy**	** *r* **	** *p* **
Main scales in AASE		
Temptation	0.57	0.000
Confidence	−0.06	0.686
Subscales of temptation in AASE		
Negative affect temptation	0.54	0.000
Social temptation	0.48	0.000
Physical temptation	0.57	0.000
Craving temptation	0.50	0.000
Confidence in AASE		
Negative affect confidence	−0.09	0.518
Social confidence	−0.03	0.817
Physical confidence	−0.03	0.840
Craving confidence	−0.07	0.625
	**Depressive Symptoms in BDI-II after Therapy**	
**Dimensions of Self-Esteem in AASE after Therapy**	** *r* **	** *p* **
Main scales in AASE		
Temptation	0.31	0.027
Confidence	0.13	0.376
Subscales of temptation in AASE		
Negative affect temptation	0.30	0.031
Social temptation	0.23	0.098
Physical temptation	0.33	0.018
Craving temptation	0.30	0.032
Confidence in AASE		
Negative affect confidence	0.16	0.277
Social confidence	0.09	0.545
Physical confidence	0.11	0.451
Craving confidence	0.15	0.305

AASE = Alcohol Abstinence Self-Efficacy Scale. BDI-II = Beck Depression Inventory Second Edition.

**Table 7 ijerph-19-09023-t007:** Standardized regression weights for associations between dimensions of self-efficacy and depressive symptoms in the investigated group before and after therapy.

	Estimate	*p*	Lower	Upper
Temptation (AASE) before therapy-Depressive symptoms (BDI-II) before therapy	0.576	0.002	0.311	0.752
Temptation (AASE) after therapy-Depressive symptoms (BDI-II) after therapy	0.306	0.009	0.063	0.506

AASE = Alcohol Abstinence Self-Efficacy Scale. BDI-II = Beck Depression Inventory Second Edition.

## Data Availability

The data presented in this study are available on request from the corresponding author. The data are not publicly available due to privacy restrictions.

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
