# Peer review of "Do Personality, Alcohol Abstinence Self-Efficacy, and Depressive Symptomatology Affect Abstinence Status in Treatment-Seeking Patients with Alcohol Use Disorder?"

_ijerph, 2022, doi:10.3390/ijerph19159023_

Round 1
Reviewer 1 Report
Congratulations on the submitted article.
Your work deals with the influence of different psychological characteristics to stop drinking alcohol. Abstinence plays an important role in the effectiveness of alcohol withdrawal. But it is not possible to find articles in the bibliography that address why some patients have willpower and others do not. Therefore, I consider that a necessary gap is covered.
It seems to me an interesting article, with potential to be published.
I believe that the number of participating individuals is reduced, as you mentioned in the limitations. A broader profile of patients would have been interesting where we would undoubtedly find more study options.
You have every argument correctly cited, and the structure of the article is clear.
I think with slight improvements it should be published. At a purely aesthetic and visual level, the tables could have another design, but I understand that something like that is subjective
Author Response
Dear Reviewer 1,
thank you for your comments. As you have mentioned, the reduced number of participants has been addressed in the Limitations section. As for the visual properties of the tables and figures, we believe that they are as clear as can be given the data they are to present. Nevertheless, thank you for this remark as well as putting in your time and effort into reviewing our work.
Reviewer 2 Report
The manuscript studied 51 inpatients that completed the International Personality Item Pool, the Alcohol Abstinence Self-Efficacy Scale and the Beck Depression Inventory and provide information on abstinence attainment 2 months post-treatment. The result showed i) a significant reduction in the severity of depressive symptoms, ii) the effect of personality and alcohol abstinence self-efficacy on depressive symptom severity, and iii) the role of personality in predicting temptation to use alcohol in recovering drinkers.
The whole study focused on different aspects based on the surveys. Have the authors considered to separate the male and female group data and reanalysis based on the different gender groups [1-2]? If female group could not generate sufficient results because of the limited numbers, would the author consider with the male groups? And did the authors consider separating the group based on the age, for example, the adult group (25-44 years old) and the middle-aged and aged group (45-62 years old).
Reference
[1] Tapert, S. F., Cheung, E. H., Brown, G. G., Frank, L. R., Paulus, M. P., Schweinsburg, A. D., ... & Brown, S. A. (2003). Neural response to alcohol stimuli in adolescents with alcohol use disorder. Archives of general psychiatry, 60(7), 727-735.
[2] Sliedrecht, W., de Waart, R., Witkiewitz, K., & Roozen, H. G. (2019). Alcohol use disorder relapse factors: A systematic review. Psychiatry Research, 278, 97-115.
Author Response
Dear Reviewer 2,
thank you for your comments and suggestions. Valid and interesting as they are, we feel that gender and age differentiation do not remain amongst the aims of this particular study. They are, however, interesting research directions that we have been considering for future analyses. That said, we would like to thank you for putting in your time and effort into reviewing our work and providing us with valuable remarks that we are most happy to use for future research endeavors.